# THINKING WITH SOUND: AUDIO CHAIN-OF-THOUGHT ENABLES MULTIMODAL REASONING IN LARGE AUDIO-LANGUAGE MODELS

## ABSTRACT

Recent Large Audio-Language Models (LALMs) have shown strong performance on various audio understanding tasks such as speech translation and Audio Q&A. However, they exhibit significant limitations on challenging audio reasoning tasks in complex acoustic scenarios. These situations would greatly benefit from the use of acoustic tools like noise suppression, source separation, and precise temporal alignment, but current LALMs lack access to such tools. To address this limitation, we introduce **Thinking-with-Sound** (TwS), a framework that equips LALMs with Audio CoT by combining linguistic reasoning with on-the-fly audio-domain analysis. Unlike existing approaches that treat audio as static input, TwS enables models to actively *think* with audio signals, performing numerical analysis and digital manipulation through multimodal reasoning. To evaluate this approach, we construct **MELD-Hard1k**, a new robustness benchmark created by introducing various acoustic perturbations. Experiments reveal that state-of-the-art LALMs suffer dramatic performance degradation on MELD-Hard1k, with accuracy dropping by more than 50% compared to clean audio. TwS achieves substantial improvements in robustness, demonstrating both effectiveness and scalability: small models gain 24.73% absolute accuracy, with improvements scaling consistently up to 36.61% for larger models. Our findings demonstrate that Audio CoT can significantly enhance robustness without retraining, opening new directions for developing more robust audio understanding systems.

## 1 INTRODUCTION

Recent advances in Large Audio-Language Models (LALMs) have enabled unified modeling of auditory and textual modalities (Tang et al., 2023; Chu et al., 2024; Défossez et al., 2024; Fang et al., 2024). Unlike traditional audio processing systems that function as task-specific solvers, LALMs allow users to specify diverse audio-related tasks through natural language instructions. This flexibility enables them to perform various audio understanding tasks including audio translation (de Seyssel et al., 2023), emotion recognition (Maimon et al., 2025), and audio Q&A (Yang et al., 2024; Wang et al., 2024). Notable examples include proprietary models like GPT-4o (OpenAI et al., 2024) and open-source contributions such as Qwen2.5 Omni (Xu et al., 2025) and Voxtral (Liu et al., 2025).

Despite these advances, current LALMs remain fundamentally limited in their acoustic understanding capabilities (Lee et al., 2025). A critical weakness lies in their limited understanding of audio signals, particularly in analyzing temporal dynamics, spectral characteristics, energy distributions, etc. The prevailing approach simply encodes audio inputs into token representations that are then processed alongside text tokens for mixed modality reasoning. While this makes good use of the language modeling capabilities of LALMs, it fundamentally constrains the models' ability to perform fine-grained acoustic analysis. The models lack mechanisms to iteratively reason about and manipulate audio in its native domain, instead treating it as a static, one-time encoded input. This architectural limitation becomes particularly pronounced when handling degraded audio or tasks requiring precise acoustic discrimination, where pure linguistic reasoning proves insufficient.

These limitations raise a critical question about how LALMs can be enhanced to reasoning with audio. Current approaches treat audio as a fixed input to be encoded once, but robust acoustic

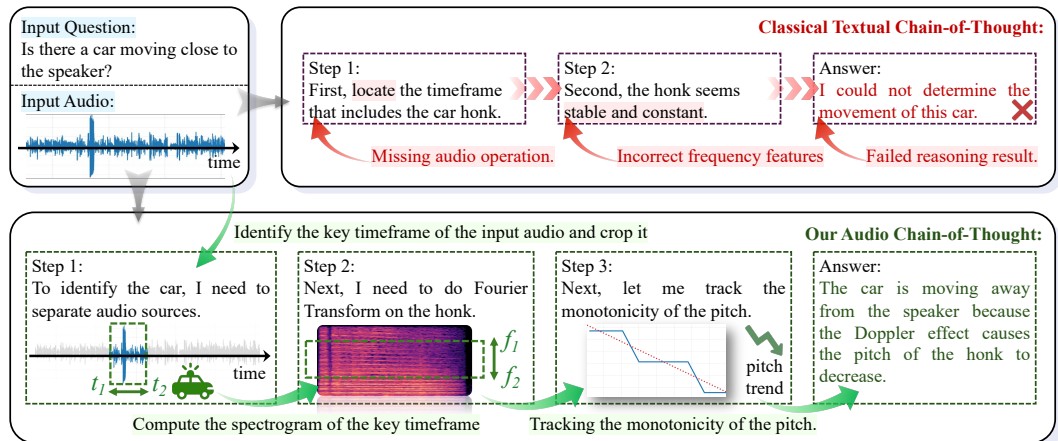

Figure 1: Our framework equips a Large Audio-Language Model with complex multimodal reasoning. Unlike traditional LALMs that struggle with acoustic details, our TwS-enabled model generates Audio Chain-of-Thought (CoT) (Wei et al., 2022) and flexibly invokes tools such as source separation and frequency analysis. This integration of linguistic reasoning with on-the-fly acoustic analysis enables accurate source identification, timestamp localization, and frequency feature extraction beyond standard inference pipelines.

understanding may require a fundamentally different paradigm. This motivates our central research question: **Can LALMs think actively with audio by iteratively analyzing and manipulating audio signals throughout the reasoning process?**

In this work, we introduce a novel **Thinking-with-Sound** reasoning framework (see Fig. 1 as overview) that enables large audio-language models (LALMs) to go beyond the limitations of purely text-based reasoning. Our approach allows the model to actively invoke appropriate tools for manipulating auditory inputs, such that the reasoning process alternates between linguistic thoughts and acoustic analysis. This design better aligns with the way humans engage in deep analysis of audio-sensitive tasks with tools, which bridges the modality gap between language and audio under complex scenarios. By jointly leveraging LALM's intrinsic reasoning capabilities and tool-augmented interactions, the model is guided to generate more coherent, reliable, and grounded multimodal chains of thought, thereby unlocking its performance bottleneck in challenging audio reasoning tasks.

For experiments, we adopt the Multimodal EmotionLines Dataset (MELD) (Poria et al., 2019) as the base benchmark and construct a new evaluation set, **MELD-Hard1k**, by introducing various types of perturbations to the audio inputs. Experimental results show that, when comparing performance on MELD and MELD-Hard1k, models of different parameter scales suffer an average accuracy drop of more than 50%. This directly highlights the substantial limitations of the zero-shot generalization ability of current LALMs. By incorporating our proposed Thinking-with-Sound (TwS) framework, we observe that even lightweight models achieve an absolute accuracy improvement of 24.73%. Moreover, as model size increases, the performance gains become more pronounced, indicating that our method amplifies the inherent audio reasoning capabilities of LALMs and demonstrates stronger generalizability and scalability.

In summary, our contributions can be summarized as follows:

(1) We propose **Thinking-with-Sound** (TwS), a novel reasoning framework that enables LALMs to perform audio CoT by interleaving linguistic reasoning with acoustic analysis.

(2) We design **MELD-Hard1k**, a robustness-oriented benchmark that introduces perturbations to systematically evaluate LALMs under challenging audio conditions.

(3) We demonstrate through extensive experiments that TwS consistently improves LALMs' gaccuracy, robustness, and scalability across model sizes, highlighting its effectiveness in unlocking the full audio reasoning capabilities of LALMs.

## 2 RELATED WORKS

**Large Audio-Language Model** LALMs represent a significant advancement beyond traditional ASR systems, enabling comprehensive audio understanding and reasoning capabilities. Recent work has explored various architectural approaches: GAMA (Ghosh et al., 2024) integrates LLMs with multiple audio representations through a custom Audio Q-Former. However, current LALMs face reliability challenges, with studies showing that even advanced models like Qwen2-Audio lack robustness awareness (Ma et al., 2025). These limitations motivate our focus on enhancing LALM reasoning through structured tool integration.

**Multimodal Chain-of-Thought** Chain-of-Thought reasoning has proven effective for complex reasoning tasks in language models (Wei et al., 2022; Kojima et al., 2022), with extensions to multimodal settings showing particular promise. Multimodal Chain-of-Thought (Zhang et al., 2023) demonstrates improved performance by incorporating vision and language modalities in a two-stage reasoning framework. Most similar to our work, Interleaved-modal Chain-of-Thought (ICoT) (Gao et al., 2025) generates sequential reasoning steps with paired visual and textual rationales, aligning more closely with human cognitive processes and significantly outperforming text-only approaches. Our work extends this paradigm from vision-language to audio-language tasks, addressing the unique challenges of temporal audio processing.

**Tool-Augmented Language Models** Integrating external tools has become central to enhancing language models. Toolformer (Schick et al., 2023) enabled autonomous API calls via self-supervision, ReAct (Yao et al., 2023) combined reasoning with tool use, and HuggingGPT (Shen et al., 2023) positioned LLMs as controllers of specialized models. In audio, MusicAgent (Yu et al., 2023) and AudioGPT (Huang et al., 2023) explored LLM-based generation, but their one-shot or pipeline designs lack the iterative refinement needed for robust understanding. Since audio is inherently temporal and sequential, effective modeling requires dynamic multi-step manipulation. Our work addresses this gap by enabling LALMs to iteratively reason over acoustic signals, refining interpretations through targeted manipulations.

## 3 METHODOLOGY

### 3.1 PROBLEM FORMULATION

We consider the setting of Large Audio-Language Models (LALMs), where the goal is to process an audio input $x_a \in \mathcal{X}$ together with a natural language instruction $x_t \in \mathcal{V}^*$ to generate a response $y \in \mathcal{V}^*$. Here, $\mathcal{X}$ denotes the space of audio signals, and $\mathcal{V}^*$ represents sequences of tokens from vocabulary $\mathcal{V}$. The response $y$ can encode various outputs including classifications, descriptions, or structured formats, depending on the task specified by $x_t$. Formally, we assume data triples $(x_a, x_t, y)$ are sampled from an underlying distribution $\mathcal{D}$, and an LALM implements a conditional distribution:

$$f_\theta(y|x_a, x_t) = \prod_{i=1}^{|y|} f_\theta(y_i|y_{<i}, x_a, x_t) \tag{1}$$

where $f_\theta$ denotes a parameterized model trained on paired audio-text data, and generation follows an autoregressive factorization. For deterministic evaluation, we consider the mode of this distribution: $y = \arg\max_{y'} f_\theta(y'|x_a, x_t)$.

### 3.2 LIMITATIONS OF CURRENT TEXT-ONLY REASONING

Current LALMs employ a one-shot encoding paradigm where the audio signal $x_a$ is compressed into a fixed sequence of embedding tokens $z_a = \text{Enc}(x_a) \in \mathbb{R}^{L \times d}$ through pre-trained audio encoders (Radford et al., 2023; Baevski et al., 2020; Hsu et al., 2021). This irreversible transformation discards fine-grained spectral and temporal information, reducing rich acoustic features to static embeddings that are then concatenated with text tokens and processed through autoregressive generation. Once encoded, the model cannot revisit the original waveform, analyze specific frequency bands, or adaptively focus on relevant temporal segments.

This architectural constraint becomes particularly limiting in scenarios requiring precise acoustic analysis. For instance, in speaker diarization tasks, the model cannot dynamically isolate and re-examine overlapping speech segments. Similarly, for emotion recognition in noisy environments, the model lacks the ability to iteratively enhance signal quality or selectively attend to emotion-bearing acoustic features like pitch contours and formant transitions. The reasoning process is thus confined to a sequence of latent states:

$$\mathcal{R} = (r_1, r_2, \ldots, r_K) \tag{2}$$

$$r_k = f_\theta(r_{<k}, z_a, x_t) \tag{3}$$

where each state $r_k$ evolves through text-space transformations without access to the underlying audio signal. Even when the model generates chain-of-thought reasoning about acoustic properties, it operates solely on the compressed representation $z_a$, unable to verify hypotheses through targeted acoustic analysis or apply corrective operations like noise suppression or temporal segmentation. This fundamental limitation—treating audio as a static input rather than a manipulable signal—constrains LALMs' ability to achieve robust understanding in challenging acoustic conditions.

### 3.3 THINKING-WITH-SOUND FRAMEWORK

We propose **Thinking-with-Sound (TwS)**, a training-free framework that augments LALMs with the ability to perform multi-step reasoning by interleaving linguistic reflection with audio-domain operations. Unlike conventional approaches that rely solely on text-based reasoning, TwS empowers models to actively manipulate and analyze audio signals during the inference process, leading to more robust and adaptive reasoning under challenging acoustic conditions.

The key insight behind TwS is that effective and human-level audio understanding often requires domain-specific operations that cannot be adequately captured even through textual level reasoning tokens alone. By allowing LALMs to invoke audio processing tools during reasoning, we enable them to: 1) Understand audio input via various acoustic tools, 2) Extract relevant features for fine-grained analysis, and 3) Iteratively refine their understanding through multi-step audio manipulation.

**General Framework.** We extend the standard reasoning process by introducing a set of audio-domain operators $\mathcal{T} = \{T_1, \ldots, T_M\}$, where each $T_m : \mathcal{X} \to \mathcal{X}$ is a transformation acting on the raw audio signal $x_a \in \mathcal{X}$. The key idea is that at each reasoning step $k$, the model can choose between two types of actions: (1) generating linguistic reasoning tokens through the LALM, or (2) applying an audio operator to transform the current audio signal. The reasoning state $r_k$ evolves by incorporating the results of both actions:

$$r_{k+1} = \begin{cases} f_\theta(r_k, \text{Enc}(x_a^{(k)}), x_t), & \phi(r_k, x_a^{(k)}, x_t) = 0, \\ f_\theta(r_k, \text{Enc}(T_m(x_a^{(k)})), x_t), & \phi(r_k, x_a^{(k)}, x_t) \neq 0 \end{cases} \tag{4}$$

where $f_\theta$ denotes the LALM's text generation function, $\text{Enc}(\cdot)$ encodes audio into token representations, and $x_t$ is the textual instruction, and $\phi(\cdot)$ is a decision function that we will be defined in the following interleaved reasoning mechanism. This formulation enables the model to iteratively refine its understanding by dynamically manipulating the audio signal based on evolving reasoning needs, rather than being constrained to a single fixed audio encoding.

**Interleaved Reasoning Mechanism** $\phi(\cdot)$ Our training-free approach leverages the inherent tool-using capabilities that existing LALMs learnt during their post-training phases. The model's decision to call an operator is formalized through:

$$d_k = \phi(r_k, x_a^{(k)}, x_t) \in \{0, 1, \ldots, M\} \tag{5}$$

where $d_k = 0$ indicates continuing linguistic reasoning and $d_k = m > 0$ indicates invoking operator $T_m$. The decision function $\phi$ represents the model's innate tool-selection capability, which evaluates the current reasoning state, audio condition, and task requirements to determine the action.

**Audio Operator Set** $\mathcal{T}$    The TwS framework is designed to be operator-agnostic, which ensures that it can adapt to arbitrary audio processing operators and domain-specific needs without architectural modifications. However, if the provided operators are irrelevant or misleading, TwS may fail to realize its full potential and, in the worst case, degenerate to the performance of the baseline method. We provide technical details in Appendix C.

**Inference Algorithm.**    The complete TwS inference procedure orchestrates the interleaved reasoning process, as detailed in Algorithm 1.

---

**Algorithm 1:** Thinking-with-Sound (TwS) Inference

**Input:** Audio $x_a$, instruction $x_t$, operators $\mathcal{T}$, LALM $f_\theta$, max steps $K_{\max}$
**Output:** Final response $y$
1  $\mathcal{R} \leftarrow \text{InitPrompt}(x_t, \mathcal{T})$
2  $k \leftarrow 0$
3  **while** $k < K_{\max}$ *and not* IsTerminated*($\mathcal{R}$)* **do**
4  |   $k \leftarrow k + 1$
5  |   $z_a \leftarrow \text{Enc}(x_a)$
6  |   $r \leftarrow f_\theta(\mathcal{R}, z_a)$ ;                          // Generate next reasoning step
7  |   **if** $m := \phi(r, x_a, x_t)$ **then**
8  |   |   $\text{args} \leftarrow \text{ParseToolCall}(r)$;
9  |   |   $x_a \leftarrow \mathcal{T}[m](x_a, \text{args})$ ;                // Apply audio transformation
10 |   $\mathcal{R} \leftarrow \mathcal{R}\|r$
11 $y \leftarrow \text{ExtractAnswer}(\mathcal{R})$
12 **return** $y$

---

This formulation captures the essential insight of TwS: the model uses its pre-trained tool-calling abilities to dynamically invoke audio operators during reasoning, creating an iterative process where linguistic analysis and audio manipulation inform each other. The framework requires no additional training and simply provides domain-specific tools that LALMs can leverage through their existing capabilities.

### 3.4 THEORETICAL ANALYSIS OF TWS

In this subsection, we will establish theoretical foundations that explain TwS's empirical effectiveness by analyzing how interleaved linguistic-acoustic mutlimodal reasoning can reduce error under perturbations.

**Preliminaries.**    Let $\mathcal{X}$ denote the raw audio signal space and we model the encoding process as $\text{Enc} : \mathcal{X} \rightarrow \mathbb{R}^{L \times d}$, which compresses audio into fixed embeddings. Given, an ideally clean audio input $x_a$ and a textual prompt $x_t$, standard LALMs generate the corresponding answer by: $y = \arg\max_o f_\theta(o|\text{Enc}(x_a), x_t)$.

For perturbed input audio signal $x_a^{\text{noisy}} = x_a + \delta$, we first formalize the error analysis:

**Definition 3.1** (Task Loss). Let $\ell : \mathcal{Y} \times \mathcal{Y} \rightarrow \mathbb{R}_+$ be a task-specific loss function. For a model $f_\theta$ with true label $y^*$, the expected loss is:

$$\mathcal{L}(x_a, x_t; f_\theta) = \mathbb{E}_{y^*}[\ell(f_\theta(\text{Enc}(x_a), x_t), y^*)] \tag{6}$$

Under the assumption that $f_\theta$ is Lipschitz continuous with constant $L_f$, we can bound the performance degradation:

$$\mathcal{L}(x_a^{\text{noisy}}, x_t; f_\theta) \leq \underbrace{\mathcal{L}(x_a, x_t; f_\theta)}_{\text{Baseline Error}} + L_f \cdot \underbrace{\|\text{Enc}(x_a^{\text{noisy}}) - \text{Enc}(x_a)\|}_{\text{Encoding Deviation}} \tag{7}$$

This decomposition separates the inherent model error on clean data from the additional error induced by acoustic perturbations through encoding differences.

**Definition 3.2** (Adaptive Operators). An operator $T \in \mathcal{T}$ is $(\epsilon, \rho)$-adaptive for perturbation type $\delta$ if for all $x_a \in \mathcal{X}$:

$$\|\delta\| \leq \epsilon \implies \|T(x_a + \delta) - x_a\| \leq \rho\|\delta\| \tag{8}$$

where $\rho < 1$ is the reduction factor. The operator set $\mathcal{T}$ is $(\epsilon, \rho)$-covering if for each perturbation type in the distribution, there exists an adaptive operator.

This definition captures the key insight: TwS succeeds when its operator set contains tools that can reduce specific perturbations encountered during inference.

**Theorem 3.3** (Error Reduction via Interleaved Reasoning). *Let $\mathcal{T}$ be an $(\epsilon, \rho)$-covering operator set with $\rho < 1$. Assume the LALM's tool selection has accuracy $\alpha > 0$ (probability of selecting an appropriate operator). After $K$ reasoning steps with TwS, let $x_a^{(K)}$ denote the processed audio. The expected encoding error satisfies:*

$$\mathbb{E}[\|Enc(x_a^{(K)}) - Enc(x_a)\|] \leq (1 - \alpha(1-\rho))^K \|Enc(x_a^{noisy}) - Enc(x_a)\| \quad (9)$$

The proof is deferred to Appendix E.1

This theorem explains the empirical observation that TwS improvements scale with model capacity: larger models have higher tool selection accuracy $\alpha$, leading to faster error reduction.

**Proposition 3.4** (Baseline Comparison). *For Lipschitz-continuous encoders (constant $L_{enc}$) and LALMs (constant $L_f$), define $L = L_f \cdot L_{enc}$. TwS with $(\epsilon, \rho)$-covering operators achieves:*

$$\mathcal{L}(x_a^{(K)}, x_t; f_\theta) \leq L \cdot (1 - \alpha(1-\rho))^K \|\delta\| + \mathcal{L}(x_a, x_t; f_\theta) \quad (10)$$

*while baseline one-shot reasoning suffers:*

$$\mathcal{L}(x_a^{noisy}, x_t; f_\theta) \leq L \cdot \|\delta\| + \mathcal{L}(x_a, x_t; f_\theta) \quad (11)$$

The proof is deferred to Appendix E.2

This formalizes why TwS recovers performance on perturbed audio while baselines fail catastrophically.

**Corollary 3.5** (Perturbation-Specific Gains). *If operator set $\mathcal{T}$ contains highly adaptive operators ($\rho \ll 1$) for perturbation type $\delta_1$ but weakly adaptive operators ($\rho \approx 1$) for $\delta_2$, then:*

$$\frac{Gain(\delta_1)}{Gain(\delta_2)} \approx \frac{1 - \rho_1}{1 - \rho_2} \quad (12)$$

The proof is deferred to Appendix E.3.

*Remark* 3.6 (Model Scaling). The tool selection accuracy $\alpha$ increases with model capacity due to improved reasoning. This creates superlinear scaling in TwS benefits: larger models both select better operators and benefit more from each operation, explaining why larger model achieves more improvements than smaller model.

These results establish that TwS's effectiveness stems from: (1) having adaptive operators, (2) the model's ability to select appropriate tools, and (3) iterative refinement that compounds improvements. In general, the framework succeeds precisely because it enables LALMs to actively analyze acoustic features that one-shot encoding pipeline cannot handle.

## 4 EXPERIMENTS

### 4.1 EXPERIMENTAL SETUP

**Benchmarks.** We evaluate TwS on emotion recognition using the Multimodal EmotionLines Dataset (MELD) (Poria et al., 2019). Additionally, to systematically evaluate robustness, we carefully curated **MELD-Hard1k** by applying controlled acoustic perturbations to 1,000 test utterances with human verification. We introduce four categories of real-world corruptions: additive noise (environmental interference), reverberation (room acoustics), pitch shifting (speaker variability), and time stretching (speech rate variations). This benchmark design allows us to isolate the impact of specific acoustic challenges while maintaining ecological validity.

**Models.** We evaluate TwS across four state-of-the-art open-source LALMs spanning different architectures and scales: Qwen2.5-Omni (3B, 7B) (Xu et al., 2025) and Voxtral (3B, 24B) (Liu et al., 2025). This selection enables assessment of TwS's generalizability across model families and its scaling properties with parameter count.

| Model | Params | MELD (Clean) | | | MELD-Hard1k (Perturbed) | | |
|-------|--------|----------|-----|----------|----------|-----|----------|
| | | Baseline | TwS | $\Delta$ | Baseline | TwS | $\Delta$ |
| Qwen2.5-Omni | 3B | 50.18 | 51.43 | +1.25 | 27.44 | 52.17 | +24.73 |
| | 7B | 47.65 | 49.21 | **+1.56** | 12.36 | 48.97 | **+36.61** |
| Audio-Flamingo3 | 7B | 48.33 | 49.81 | +1.48 | 18.71 | 50.16 | +31.45 |
| Voxtral | 3B | 44.95 | 45.38 | +0.43 | 30.05 | 41.43 | +11.38 |
| | 24B | 51.62 | 53.14 | **+1.52** | 24.55 | 49.49 | **+24.94** |

Table 1: Performance comparison of baseline LALMs versus TwS-enhanced models on clean (MELD) and perturbed (MELD-Hard1k) audio. $\Delta$ denotes absolute accuracy gain. Best performances among the s*ame model architecture* are highlighted in **bold**.

**Configuration and Metrics.** For TwS implementation, we configure the framework with a maximum of $K_{\max} = 5$ reasoning steps. Our training-free approach ensures fair comparison with baseline models while leveraging LALMs' inherent tool-using capabilities. We measure emotion classification accuracy as our primary metric, comparing baseline LALM performance against TwS-enhanced models on both clean (MELD) and perturbed (MELD-Hard1k) conditions.

See Appendix A for more implementation details.

## 4.2 MAIN RESULTS

Table 1 presents our main experimental results comparing baseline performance against our TwS method on both clean (MELD) and perturbed (MELD-Hard1k) audio conditions. We evaluate four state-of-the-art LALMs spanning different architectures and scales to assess the generalizability and scalability of our approach.

On the original MELD dataset, baseline models achieve emotion recognition accuracies ranging from 44.95% (Voxtral-3B) to 51.62% (Voxtral-24B). When TwS is applied to clean audio, we observe improvements of 0.43-1.56 percentage points, demonstrating that our framework enhances reasoning even when audio quality is not the primary limiting factor.

While these improvements on clean audio are modest, the true value of TwS becomes apparent when examining performance on MELD-Hard1k, where acoustic perturbations reveal critical vulnerabilities in current LALMs. All baseline models experience substantial performance degradation, with accuracy drops exceeding 50% relative to their clean performance. The most severe case, Qwen-7B, declines from 47.65% to 12.36%. In contrast to these baseline failures, TwS demonstrates strong effectiveness in handling perturbations, achieving absolute accuracy gains ranging from 11.38 percentage points (Voxtral-3B) to 36.61 percentage points (Qwen-7B). The framework's recovery capabilities are particularly striking: TwS-enhanced models on perturbed audio often approach or exceed their baseline performance on clean audio, effectively compensating for acoustic corruptions. For instance, Qwen-3B with TwS achieves 52.17% on MELD-Hard1k, surpassing its own baseline performance of 50.18% on clean audio.

Beyond these individual improvements, our results reveal an intriguing pattern in how TwS's effectiveness scales with model size. While larger models generally achieve better baseline performance on clean audio, they are not necessarily more robust to perturbations (Qwen-7B retains only 25.9% of its clean performance under perturbation, compared to 54.7% for Qwen-3B). However, the effectiveness of TwS correlates positively with model capacity, with relative improvements on MELD-Hard1k increasing from 37.9% for Voxtral-3B to 101.6% for Voxtral-24B, and even more pronounced scaling in Qwen series (90.1% for 3B versus 296.3% for 7B). This pattern suggests that larger models can better leverage the structured reasoning process enabled by TwS, potentially due to their enhanced capacity to coordinate between linguistic reasoning and audio manipulation. The superlinear scaling of improvements with model size indicates that TwS unlocks latent audio reasoning capabilities that were previously underutilized in standard inference pipelines.

## 4.3 ABLATION STUDIES

To understand the mechanisms underlying TwS's effectiveness, we conduct systematic ablation studies examining the contribution of operators, reasoning dynamics, and computational trade-offs.

**Operator Contribution Analysis.** Although TwS is operator-agnostic, we evaluate one instantiation with four operator categories: denoising, enhancement, normalization, and analysis. These categories, chosen for their relevance to our tasks, illustrate the framework's effectiveness. Table 2 reports leave-one-out results. Denoising proves most critical, with its removal causing a 15.8% absolute accuracy drop, consistent with the prevalence of additive noise in MELD-Hard1k. Enhancement yields a 7.2% gain, particularly for temporal distortions. Normalization offers modest but consistent improvements (3.4%), while analysis mainly supports subsequent operator selection rather than direct transformation. These results reflect our chosen operators and benchmarks; alternative sets would likely show different patterns while preserving the principle of adaptive tool selection.

| Configuration | Denoise | Enhance | Normalize | Analyze | Accuracy (%) | $\Delta$ |
|---|---|---|---|---|---|---|
| TwS (our) | ✓ | ✓ | ✓ | ✓ | 48.97 | — |
| w/o Denoising | × | ✓ | ✓ | ✓ | 33.17 | $-15.80$ |
| w/o Enhancement | ✓ | × | ✓ | ✓ | 41.77 | $-7.20$ |
| w/o Normalization | ✓ | ✓ | × | ✓ | 45.57 | $-3.40$ |
| w/o Analysis | ✓ | ✓ | ✓ | × | 47.23 | $-1.74$ |
| Baseline | × | × | × | × | 12.36 | $-36.61$ |

Table 2: Operator ablation study on MELD-Hard1k. Each row removes one operator category while retaining others. ✓ indicates the operator category is included, × indicates removal.

**Reasoning Dynamics.** Figure 2 reveals the relationship between maximum allowed reasoning steps and performance. Most samples converge within 3-4 steps, with diminishing returns beyond $K_{\max} = 5$. Interestingly, the average number of steps used, 2.8, is substantially lower than the maximum, indicating that the model has the ability to terminate reasoning once sufficient confidence is achieved. The computational overhead scales linearly with steps used, suggesting that adaptive early stopping provides an effective efficiency-accuracy trade-off.

**Perturbation-Specific Performance.** To understand where TwS provides the greatest benefits, we analyze performance across different perturbation types in Figure 3. As shown in Figure 3(b), TwS demonstrates remarkable effectiveness against additive noise (+35.2%) and reverberation (+28.7%), where targeted operators can directly address these corruptions. Pitch shift sees moderate improvements (+18.3%), primarily through frequency-domain adjustments. Time stretching proves most challenging, with only 12.1% improvement, as temporal distortions fundamentally alter phonetic patterns that are difficult to recover through signal processing alone.

The operator usage patterns depicted in Figure 3(a) align with intuition: noise-targeted operators dominate for noise corruption (68% of invocations), while enhancement operators are preferentially selected for time-stretched audio

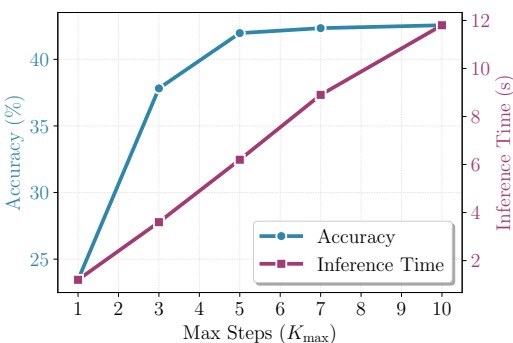

Figure 2: Impact of maximum reasoning steps on performance and efficiency. Inference time measured on NVIDIA A100 GPU, averaged over 100 samples. The figure shows accuracy (left y-axis) and inference time (right y-axis) as functions of maximum allowed reasoning steps.

(45% of invocations). For pitch-shifted audio, frequency-adjustment operators take precedence (42%), reflecting their natural alignment with this perturbation type. The consistent usage of analysis operators (10-13% across all perturbations) indicates the model's systematic approach to understanding corruption characteristics before applying corrective measures, validating TwS's adaptive reasoning mechanism.

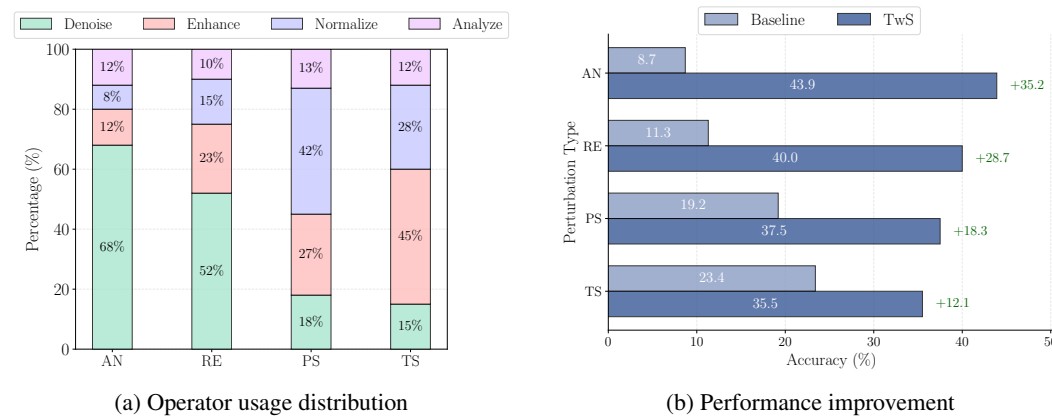

(a) Operator usage distribution        (b) Performance improvement

Figure 3: Performance breakdown by perturbation type (AN = Additive Noise, RE = Reverberation, PS = Pitch Shift, TS = Time Stretch). (a) Operator usage distribution across perturbations; (b) Accuracy comparison between baseline and TwS, with improvement percentages annotated. TwS shows aligned operator usage rate and consistent improvements across different perturbation types.

## 5 DISCUSSION

### 5.1 WHY DOES TwS WORK?

The effectiveness of TwS stems from its ability to enable multimodal reasoning, where models actively *think with audio*, thereby addressing a critical limitation of current LALMs' naive Chain-of-Thought. Specifically, TwS supports an audio CoT that enables LALMs to perform precise audio signal processing operations, interleaved cross-modal reasoning, and iterative refinement during problem solving. Importantly, the improvements of TwS scale with model capacity, indicating that larger models can more effectively coordinate interleaved reasoning that bridge acoustic observations with linguistic reasoning under our framework.

### 5.2 COMPUTATIONAL TRADE-OFFS

TwS improves accuracy at the cost of higher inference overhead, mainly from additional reasoning steps. Most samples converge within 2–4 iterations with minimal latency from audio operators (Fig. 2). On Qwen-7B, inference is about $2.3\times$ slower than naive CoT. Larger models require fewer steps yet yield greater gains, suggesting favorable scaling. For real-time use, adaptive stopping or confidence-based thresholds can further mitigate latency.

## 6 CONCLUSION

We introduced Thinking-with-Sound (TwS), a training-free framework that enables Large Audio-Language Models to perform multi-step reasoning by interleaving linguistic analysis with dynamic audio manipulation. Unlike existing approaches that treat audio as static input, TwS allows models to iteratively process and re-examine acoustic signals, addressing the fundamental limitation that current LALMs cannot perform fine-grained acoustic analysis despite their strong linguistic capabilities. Our experiments on MELD-Hard1k demonstrate that while state-of-the-art LALMs suffer catastrophic performance degradation under acoustic perturbations (>50% accuracy drop), TwS achieves substantial recovery with improvements ranging from 24.73% to 36.61% absolute accuracy, scaling with model capacity. These results, supported by theoretical analysis establishing expressive completeness and robustness guarantees, demonstrate that effective audio understanding requires reasoning through acoustic signals rather than merely reasoning about them. By enabling models to actively manipulate audio during inference, TwS provides a practical path toward more robust audio-language systems with multimodal reasoning.

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

## A  IMPLEMENTATION DETAILS

### A.1  DATASETS

The base dataset, MELD, contains 13,708 utterances from conversational contexts with seven emotion categories. MELD's naturalistic audio conditions, including overlapping speech, background noise, and varied prosody, provide an ideal testbed for assessing LALMs' acoustic reasoning capabilities beyond clean laboratory conditions. See Appendix B for **MELD-Hard1k**.

### A.2  HYPER-PARAMETERS

We use NVIDIA A100 GPUs with fixed random seeds (seed=42 for sampling, seed=1337 for perturbations). Model inference employs default parameters (temperature=0, top-p=0.95) with greedy decoding for deterministic evaluation. Complete implementation including perturbation generation, TwS framework, and evaluation scripts will be released upon publication.

## B  PERTURBATION CONFIGURATION

We use the following perturbation configuration (see Table. 3) when constructing the **MELD-Hard1k** dataset.

Table 3: Detailed perturbation specifications for MELD-Hard1k construction. Each perturbation type is applied with probability $p = 0.3$, with parameters sampled uniformly from the specified ranges.

| Pert. Type | Parameter | Range | Dist. | Impl. |
|---|---|---|---|---|
| Additive Noise | SNR (dB) | [0, 25] | Uniform | $x' = x + \alpha \cdot n(t)$ |
| | Noise Type | {white, pink, brown} | Categorical | |
| | Temporal Mask | [0, 1] | Bernoulli(0.2) | |
| Reverberation | RT60 (ms) | [100, 800] | Log-uniform | $x' = x * h_{room}(t)$ |
| | Room Size (m³) | [20, 200] | Uniform | |
| Pitch Shift | Semitones | [-4, +4] | Uniform | PSOLA algorithm |
| | Formant Pres. | {True, False} | Bernoulli(0.7) | |
| Time Stretch | Stretch Fact. | [0.7, 1.3] | Uniform | Phase vocoder |
| | Quality Mode | {fast, high} | Bernoulli(0.8) | |

## C  DESIGN OF AUDIO OPERATOR SET $\mathcal{T}$

While TwS imposes no hard constraints on the operator set, our empirical analysis highlights consistent patterns in what makes operators effective for audio reasoning. Operators that facilitate strong performance typically share three characteristics: (1) they implement functionalities that LALMs are not inherently good at, such as frequency-domain analysis and pitch tracking tasks which require accurate numerical operation / analysis. (2) they return required data directly, without additional descriptive text; and (3) they are documented with precise specifications and clear boundaries, including intuitive names and well-defined parameters, so the agent can reliably determine when and how to invoke them.

In our experiments, for example, we instantiate $\mathcal{T}$ with operators spanning enhancement (denoising, echo cancellation), analysis (spectral analysis, pitch tracking), transformation (time-frequency manipulations), and separation (source separation, human voice extraction). This particular choice reflects common audio reasoning needs in our evaluation but LALMs are not natively good at.

Nonetheles, our TwS framework naturally accommodates alternative operator sets. For instance, speech recognition tasks might prioritize formant enhancement and silence removal, while music analysis could benefit from harmonic-percussive separation and beat tracking—the same TwS framework applies regardless of the specific operators employed.

# D    PROMPTS

To ensure reproducibility, we provide the complete prompts used in our experiments. We employed two main categories of prompts: baseline prompts for standard LALM evaluation and TwS-enhanced prompts that enable audio chain-of-thought reasoning with tool integration.

## D.1    BASELINE EVALUATION PROMPTS

For baseline experiments, we used standard emotion recognition prompts without any tool-calling capabilities.

---
**Baseline Prompt**

**Instruction:** You are an expert in audio analysis and emotion recognition. Listen to the provided audio clip and identify the speaker's emotional state.
The audio contains a single speaker's utterance from a conversational context. Your task is to classify the emotion expressed in the speech.
Choose from the following emotion: {categories:,anger,disgust,fear,joy,neutral,sadness,surprise}
Think step-by-step and provide your answer in the following format: Emotion: [category]
[*AUDIO INPUT*]

---

Figure 4: The baseline prompt used for standard LALM emotion recognition evaluation.

## D.2    TwS FRAMEWORK PROMPTS

The TwS framework requires more sophisticated prompts that introduce tool-calling capabilities and guide the model through multi-step audio reasoning processes.

---
**TwS System Prompt**

**System Instruction:** You are an advanced audio analysis system with access to specialized audio processing tools. Your goal is to perform comprehensive emotion recognition by actively analyzing and manipulating audio signals when needed.
You have access to the following audio processing tools:
[Detailed Operator Set Description]
When encountering audio that may be degraded or challenging to analyze, think step-by-step:
1. First assess the audio quality and identify potential issues
2. Apply appropriate preprocessing or enhancement tools as needed
3. Perform detailed acoustic analysis using available tools
Format tool calls as:
[TOOL: tool_name(parameters)]

---

Figure 5: The system prompt that initializes TwS framework capabilities and introduces available audio processing tools.

# E    PROOFS

## E.1    PROOF OF THEOREM 3.3

*Proof.* We analyze the error evolution over reasoning steps. At step $k$, let $x_a^{(k)}$ denote the current audio state. If the model selects an appropriate operator $T$ (which occurs with probability $\alpha$), we have:

$$\|x_a^{(k+1)} - x_a\| = \|T(x_a^{(k)}) - x_a\| \tag{13}$$

$$\leq \rho\|x_a^{(k)} - x_a\| \quad \text{(by } (\epsilon, \rho)\text{-adaptivity)} \tag{14}$$

If the model continues linguistic reasoning (probability $1 - \alpha$), the audio remains unchanged: $\|x_a^{(k+1)} - x_a\| = \|x_a^{(k)} - x_a\|$.

---

**TwS Task Prompt**

**Task Instruction:** Analyze the provided audio clip to determine the speaker's emotional state. Use your available tools strategically to ensure accurate analysis, especially if the audio quality presents challenges.
Emotion categories: {anger,disgust,fear,joy,neutral,sadness,surprise}
Process:
1. Initial Assessment: Listen to the audio and evaluate its quality
2. Strategic Processing: If needed, apply appropriate tools to enhance or analyze the audio
3. Feature Extraction: Use analysis tools to extract emotion-relevant acoustic features
4. Integration: Combine your observations to reach a conclusion
5. Final Decision: Provide emotion classification.
Think through each step explicitly. Show your reasoning process and explain how each tool usage contributes to your final decision.

Expected output format:
Step-by-step Analysis: [Your detailed reasoning process with tool calls]
Final Answer:
Reasoning: [brief justification]
Emotion: [category]
[*AUDIO INPUT*]

---

Figure 6: The task-specific prompt used for TwS-enhanced emotion recognition, guiding multi-step reasoning and tool usage.

Taking expectations over the model's stochastic tool selection:

$$\mathbb{E}[\|x_a^{(k+1)} - x_a\|] = \alpha \cdot \rho \|x_a^{(k)} - x_a\| + (1 - \alpha) \cdot \|x_a^{(k)} - x_a\| \tag{15}$$

$$= (1 - \alpha(1 - \rho))\|x_a^{(k)} - x_a\| \tag{16}$$

Unrolling this recursion from $k = 0$ to $K$:

$$\mathbb{E}[\|x_a^{(K)} - x_a\|] \leq (1 - \alpha(1 - \rho))^K \|x_a^{(0)} - x_a\| \tag{17}$$

Since the encoding is Lipschitz (or at least continuous), this bound on audio-space error translates to the encoding-space error bound in the theorem statement. $\square$

### E.2 PROOF OF PROPOSITION 3.4

*Proof.* For TwS, after $K$ steps with error reduction from Theorem 3.3:

$$\mathcal{L}(x_a^{(K)}, x_t; f_\theta) \leq \mathcal{L}(x_a, x_t; f_\theta) + L_f \cdot \|\text{Enc}(x_a^{(K)}) - \text{Enc}(x_a)\| \tag{18}$$

$$\leq \mathcal{L}(x_a, x_t; f_\theta) + L_f \cdot L_{\text{enc}} \cdot \|x_a^{(K)} - x_a\| \tag{19}$$

$$\leq \mathcal{L}(x_a, x_t; f_\theta) + L \cdot (1 - \alpha(1 - \rho))^K \|\delta\| \tag{20}$$

where $L = L_f \cdot L_{\text{enc}}$ combines the Lipschitz constants of the model and encoder.

For baseline one-shot reasoning without TwS:

$$\mathcal{L}(x_a^{\text{noisy}}, x_t; f_\theta) \leq \mathcal{L}(x_a, x_t; f_\theta) + L_f \cdot \|\text{Enc}(x_a^{\text{noisy}}) - \text{Enc}(x_a)\| \tag{21}$$

$$\leq \mathcal{L}(x_a, x_t; f_\theta) + L \cdot \|\delta\| \tag{22}$$

The improvement factor is $(1 - \alpha(1 - \rho))^K < 1$, showing TwS strictly reduces error when operators are adaptive ($\rho < 1$) and the model can select them ($\alpha > 0$). $\square$

### E.3 PROOF OF COROLLARY 3.5

*Proof.* The gain from TwS for perturbation type $\delta_i$ with reduction factor $\rho_i$ is:

$$\text{Gain}(\delta_i) = \mathcal{L}_{\text{baseline}}(\delta_i) - \mathcal{L}_{\text{TwS}}(\delta_i) \approx L\|\delta_i\| \left(1 - (1 - \alpha(1 - \rho_i))^K\right) \tag{23}$$

For similar perturbation magnitudes $\|\delta_1\| \approx \|\delta_2\|$ and moderate $K$, taking the ratio:

$$\frac{\text{Gain}(\delta_1)}{\text{Gain}(\delta_2)} \approx \frac{1 - (1 - \alpha(1 - \rho_1))^K}{1 - (1 - \alpha(1 - \rho_2))^K} \tag{24}$$

$$\approx \frac{\alpha K(1 - \rho_1)}{\alpha K(1 - \rho_2)} = \frac{1 - \rho_1}{1 - \rho_2} \tag{25}$$

where we used the approximation $(1 - x)^K \approx 1 - Kx$ for small $x$. $\qquad\square$

## F  THE USE OF LARGE LANGUAGE MODELS (LLMS)

We used an LLM to assist with the phrasing and grammar of the manuscript. The LLM was used strictly as a writing aid and did not contribute to the scientific ideation, methodology, or results presented in this paper.

