# OpenReview forum: "Thinking with Sound: Audio Chain-of-Thought Enables Multimodal Reasoning for Large Audio-Language Models"
_ICLR.cc/2026/Conference — ICLR 2026 Conference Withdrawn Submission_

### Official Review · Reviewer_WmBt · 2025-10-31

**Soundness:** 3
**Presentation:** 3
**Contribution:** 2
**Rating:** 4
**Confidence:** 4

**Summary:**

This paper proposes the Thinking-with-Sound (TwS) framework, which introduces the Audio Chain-of-Thought (Audio CoT) to LALMs. The TwS framework enables models to actively analyze and **manipulate** audio signals using tools during the reasoning process, alternating between linguistic reasoning and acoustic domain operations (such as noise reduction, pitch tracking, temporal segmentation, etc.). Unlike traditional LALMs that only treat audio as static embeddings, TwS can dynamically call tools to perform multi-step audio reasoning. This enhances the LALMs' ability to understand corrupted audio signals **without additional training**. Additionally, the authors constructed a new robustness benchmark, **MELD-Hard1k**, which evaluates models' noise resistance by applying various real-world perturbations to audio. Experimental results show that TwS significantly improves the robustness of multiple LALMs without extra training, with a maximum performance gain of 36.61%. Furthermore, the theoretical analysis section provides an explanation for the error convergence of TwS under perturbed conditions.

**Strengths:**

The authors argue that TwS can significantly enhance the robustness and reasoning capabilities of LALMs without the need for retraining. Experimental results (Table 1) show that TwS achieves a substantial improvement in accuracy on distorted audio; Ablation studies (Table 2) indicate that **noise reduction** and **enhancement operators** contribute the most; And reasoning step analysis (Figure 2) proves that TwS converges with a small number of iterations while maintaining good computational efficiency.

The theoretical section (Section 3.4) formally analyzes the mechanism of TwS from the perspective of coding errors, and demonstrates that alternating reasoning and adaptive operators can reduce perturbation errors. The overall derivation logic is clear and consistent with experimental results.

**Weaknesses:**

1. The theoretical section (Section 3.4) relies on strong assumptions (such as the Lipschitz continuity of the encoder and the stability of tool selection accuracy α)**, so its generalization still needs to be verified in more empirical studies.

2. Notably, although the results on MELD-Hard1k fully demonstrate the improvement in robustness, the evaluation scope is **limited to emotion recognition tasks** and does not yet cover more complex audio understanding or question-answering tasks.

3. The results are theoretically plausible and align with the empirical findings, but future work could test these assumptions empirically.

4. The paper extends multimodal chain-of-thought reasoning (Zhang et al., 2023; Gao et al., 2025) to the audio domain and connects to tool-augmented reasoning (Toolformer, ReAct, HuggingGPT). The novelty lies in unifying these paradigms under a “training-free” audio manipulation framework.

5. The experiment lacks sufficient breadth, with its covered tasks and models being too limited. The authors could attempt to enrich the experimental tasks and include more audio-language large models (LALMs).

**Questions:**

1. In Appendix C, the authors primarily adhere to two core logics when selecting operators: "addressing the shortcomings of LALMs" and "adapting to specific task scenarios". I am quite confused about how these tools fulfill their corresponding requirements. For example, how does denoising work? Suppose I blindly pass an audio sample through all the tools, then input it to the LALM and ask it to answer (without TwS). How would its performance change compared to that in this paper?

---

### Official Review · Reviewer_tKyS · 2025-11-01

**Soundness:** 2
**Presentation:** 3
**Contribution:** 2
**Rating:** 2
**Confidence:** 5

**Summary:**

The paper introduces Thinking-with-Sound (TwS), a training-free framework that enhances Large Audio-Language Models (LALMs) by enabling an interleaving of linguistic reasoning with active audio signal manipulation during inference. To test robustness, the authors construct MELD-Hard1k, a benchmark with controlled audio perturbations (noise, reverberation, pitch shift, time stretch). Experiments across multiple LALM architectures show that TwS recovers 24–37% absolute accuracy, with gains scaling positively with model size. Ablations show denoising as most impactful, and perturbation-specific analysis reveals TwS handles noise/reverberation best.

**Strengths:**

1. Clear identification of a key limitation in current LALMs, the inability to re-access/manipulate raw audio during reasoning.
2. Detailed ablations (operator contribution, reasoning step dynamics, perturbation-specific breakdown) that give insight into where and why TwS works.
3. The quality of the writing is good.

**Weaknesses:**

1. Narrow task and evaluation scope. The evaluation is limited to MELD emotion recognition, which primarily contains speech data. There is no testing on broader, widely used standard multimodal audio benchmarks, MMAU[1] or MMAR[2], which would provide a more comprehensive assessment. The current setup lacks evaluation involving music, environmental sounds, or more complex multi-step reasoning tasks, limiting the generalizability of the conclusions to the full spectrum of audio-language understanding.
2. Missing key related work. The paper does not cite recent closely related studies such as Audio-CoT[3] and Audio-Reasoner[4]. These works are pioneers in doing CoT for LALM, which share conceptual similarities with TwS in enabling step-by-step audio reasoning. Omitting them weakens the positioning and novelty claims.
3. Questionable necessity of the theoretical analysis in Section 3.4. The error analysis for “Interleaved reasoning with tool calling” is largely modality-agnostic and essentially applies the same reasoning framework used in image-text models with tool integration. The only differences lie in the modality-specific signals and operator sets. This section could be moved to the appendix to streamline the main paper and focus on the core method and empirical findings.

[1] MMAU: A Massive Multi-Task Audio Understanding and Reasoning Benchmark

[2] MMAR: A Challenging Benchmark for Deep Reasoning in Speech, Audio, Music, and Their Mix

[3] Audio-CoT: Exploring Chain-of-Thought Reasoning in Large Audio Language Model

[4] Audio-Reasoner: Improving Reasoning Capability in Large Audio Language Models

**Questions:**

See Weaknesses. The main concern is that the evaluation is too weak. In the current landscape, widely adopted benchmarks for audio understanding (MMAU) and audio reasoning (MMAR) are already available, yet the paper chooses not to use them and instead builds its own emotion recognition task to demonstrate the effectiveness of Thinking-with-Audio. This is not very convincing. Without using established benchmarks, it is difficult to determine whether the proposed method is truly effective.

---

### Official Review · Reviewer_anZo · 2025-11-01

**Soundness:** 2
**Presentation:** 2
**Contribution:** 3
**Rating:** 4
**Confidence:** 4

**Summary:**

An interesting paper that proposes agentic  behavior facilitated by the use of tools to get prompt specific features from the audio being reasoned about via tool calling. Results are quite impressive in the limited evaluation.

There is an attempt to provide mathematical rigor, but I believe their analysis is flawed, and does not contribute to the results. Moreover, he authors seem unaware of literature around standards such as MCP facilitating tool calling.

**Strengths:**

Tools such as web search, date check etc., have become common in agentic frameworks via protocols such as MCP (which the authors do not refer to!), and it would be interesting to extend to more signal processing based tools for audio, which this paper does. Results on the three models are quite good

**Weaknesses:**

I would have liked to see audio samples of the tool operations used with the paper, and also seen in the appendix a list of the decision the model made in solving a particular task.

Another question I have is if the perturbations introduced in the audio and the operators are effectively drawn from the same family, allowing for an easy reverse application and clean up?

**Questions:**

The proof of theorem 3.3 needs more careful consideration. For example, in equation 15, the left side is deterministic, and the right side is not. This needs to be written out more rigorously. Moreover, it is unclear how the result in the audio space extends to the encoded space. The Lipschitz inequality provides half of the argument but just using it will not be enough:
E[||Enc(x^k_a)-Enc(x_a)||] \leq L E || x^k_a -x_a|| \leq L E|| x^0_a-x_a|| \leq ??
 How do you get to the encoded version from here?

---

### Official Review · Reviewer_j535 · 2025-11-10

**Soundness:** 3
**Presentation:** 3
**Contribution:** 3
**Rating:** 4
**Confidence:** 4

**Summary:**

The paper introduces Thinking With Sound which is a training-free framework which lets large audio language models introduce linguistic reasoning with on the fly audio operations (e.g., source separation, denoising, spectral analysis) using an Audio Chain-of-Thought. Unlike conventional approaches that rely solely on text-based reasoning, TwS empowers models to actively manipulate and analyze audio signals during the inference process, leading to more robust and adaptive reasoning under challenging acoustic conditions. The paper also introduces MELD-Hard1k, a benchmark set created by perturbing MELD utterances with additive noise, reverberation, pitch shift, and time-stretch. The paper conducts experiments to show that TwS improves LALMs’ accuracy, robustness, and scalability across different model sizes.

**Strengths:**

- Introduces thinking with audio instead of thinking about audio using audio operators
- robustness gains across different models using the technique
- the framework is operator-agnostic, which ensures that it can adapt to arbitrary audio processing operators
- presents a novel audio operator benchmark

**Weaknesses:**

- Results are limited to emotion classification on MELD/Hard1k, it’s unclear how TwS transfers to other audio-reasoning tasks (spatial, event, multi-clip)
- Missing discussion of failure cases/ wrong tool calls (qualitative analysis). Where are models going wrong?
- Wouldn't Operator Contribution Analysis make more sense if done on an independent set? Currently the results will reflect the distribution of test set created?
- (nit) line 321 four -> five
- The paper introduces α (tool-selection accuracy) and ρ (operator adaptivity) in its theory but does not calculate or estimate either of them in the experiments. Need a section to estimate the varialbes

**Questions:**

- I understand that the perturbations were applied with a probability of 0.3 but what's the distribution of perturbations in the actual MELD-Hard1k set?
- With TwS 2.3× slower on Qwen-7B, how do results change under a fixed inference time budget?

---

### Note · Authors · 2025-12-01

I have read and agree with the venue's withdrawal policy on behalf of myself and my co-authors.